# Internet of Things: A Review on Theory Based Impedance Matching Techniques for Energy Efficient RF Systems

Benoit Couraud [1,*], Remy Vauche [2], Spyridon Nektarios Daskalakis [1], David Flynn [1], Thibaut Deleruyelle [2,3], Edith Kussener [2,3] and Stylianos Assimonis [4]

[1] Smart Systems Group, Heriot-Watt University, Edinburgh EH14 4AS, UK; daskalakispiros@gmail.com (S.N.D.); D.Flynn@hw.ac.uk (D.F.)
[2] Aix Marseille University, Université de Toulon, CNRS, IM2NP, 13397 Marseille, France; remy.vauche@univ-amu.fr (R.V.); thibaut.deleruyelle@yncrea.fr (T.D.); edith.kussener@yncrea.fr (E.K.)
[3] ISEN-Toulon, CNRS, IM2NP, 83000 Toulon, France
[4] Institute of Electronics, Communications and Information Technology (ECIT), Queen's University Belfast, Belfast BT3 9DT, UK; S.Assimonis@qub.ac.uk
\* Correspondence: b.couraud@hw.ac.uk

**Abstract:** Within an increasingly connected world, the exponential growth in the deployment of Internet of Things (IoT) applications presents a significant challenge in power and data transfer optimisation. Currently, the maximization of Radio Frequency (RF) system power gain depends on the design of efficient, commercial chips, and on the integration of these chips by using complex RF simulations to verify bespoke configurations. However, even if a standard $50\,\Omega$ transmitter's chip has an efficiency of 90%, the overall power efficiency of the RF system can be reduced by 10% if coupled with a standard antenna of $72\,\Omega$. Hence, it is necessary for scalable IoT networks to have optimal RF system design for every transceiver: for example, impedance mismatching between a transmitter's antenna and chip leads to a significant reduction of the corresponding RF system's overall power efficiency. This work presents a versatile design framework, based on well-known theoretical methods (i.e., transducer gain, power wave approach, transmission line theory), for the optimal design in terms of power delivered to a load of a typical RF system, which consists of an antenna, a matching network, a load (e.g., integrated circuit) and transmission lines which connect all these parts. The aim of this design framework is not only to reduce the computational effort needed for the design and prototyping of power efficient RF systems, but also to increase the accuracy of the analysis, based on the explanatory analysis within our design framework. Simulated and measured results verify the accuracy of this proposed design framework over a 0–4 GHz spectrum. Finally, a case study based on the design of an RF system for Bluetooth applications demonstrates the benefits of this RF design framework.

**Keywords:** Internet of things (IoT); RF circuit; RF integration; transmission line theory

## 1. Introduction

The number of installed Internet of Things (IoT) endpoints could reach 50 billions units in 2030 [1]. The demand that is driving this global growth of IoT innovation includes factors such as a change in consumer trends and in expectations of technology, increasing growth of global operations requiring data driven design and control of distributed assets, systems and networks [2,3]. Along with these trends, global policy is also shaping trends in IoT as countries look to implement circular economy and decarbonisation initiatives, with IoT providing improved knowledge and visibility of the performance of systems [4–6]. IoT is a fundamental enabler to new, globally competitive services, as companies seek to grow in response to evolving technology and consumer demands. For example, bidirectional and interactive information and engagement with distributed systems (assets) via internet connectivity. Emergent and disruptive technologies that will influence society as well as

industry, such as autonomous systems and artificial intelligence (AI), are also dependent on IoT for operational and planning functions in society and industry [7]. Without IoT operational performance and safety compliance, constraints would inhibit the growth of these technologies. Overall, the ultimate need for industry to drive IoT is to enhance operational efficiency, mitigate risks, improve functional visibility, provide maximum customer engagement, increase revenue streams, and remove barriers to entry into new market opportunities for growth. The underlying enabler to all of this is real-time information and insights provided by IoT and connected devices. Hence, the success of the increase in IoT device deployment throughout a myriad of applications has actually been due to the affordability, ease of integration and use of this technology [8–10]. Indeed, the rate of adoption of IoT into various markets is based on its fundamental ability to easily scale and be customisable based on the integration of affordable and readily available technology. One of the main challenges is to minimize the cost and energy consumption of the existing IoT devices. There is a variety of wireless sensor products in the market (i.e., ZigBee, LoRa) from 40 to 400 USD per sensor-node [11]. Thus, the networking cost of 100 plants (e.g., one sensor/plant) in precision agriculture, for example, becomes prohibitive. One solution on this problem is a novel technique, based on reflection principles and its backscatter communication [12–14]. However, to make this growth sustainable in terms of energy consumption and cost, it is necessary to ensure optimal energy efficient designs of these devices. Similarly, other recent Radio Frequency (RF) applications also require efficient RF designs, such as Wireless Power and Data Transfer (WPDT), energy harvesting [15–23], or any energy efficient applications such as Bluetooth Low Energy, LoRa and Sigfox.

Such efficient designs require a system approach from which each component that constitutes an IoT system will be carefully designed [24], as shown in Figure 1. Current research works in circuits and systems' efficiency mostly focus on the design of efficient protocols or architectures for communication Integrated Circuit (IC) [25–28], or on more efficient power management systems [29–31], but they usually do not include efficient integration of the proposed IC into a comprehensive IoT system. In this way, the design of the front end IC usually aims to meet an impedance of 50 Ω, such that the design of the transmission line and the matching circuit or antenna can be reduced to the choice of components with characteristic impedance equal 50 Ω. However, bespoke designs could be achieved if a system approach was used that includes the impedance matching aspects [24,32,33]. Indeed, today's packaging constraints often prevent engineers to use standard components with specific sizes, leading to complex computer aided (CAD) design methods or to inefficient designs that prevent scalability of IoT networks. Hence, some works have proposed theory-based impedance matching guidelines in a general context [34–37] or for specific applications, such as dual-band matching [38] or adaptative impedance matching [39,40]. Similarly, engineering solutions to improve RF systems' efficiency usually uses linear matching networks with lumped elements and elementary transmission lines elements done by means of Smith Chart techniques. However, these works make it difficult to address the design of more complex geometries with packaging constraints if not using RF CAD software. This article proposes a design methodology to quickly design transmission lines and matching network for typical RF systems, such as the one shown in Figure 2, constituted by a source with an impedance $Z_g$ (e.g., antenna), a given load (e.g., IC), a matching network in between them and transmission lines, which electromagnetically connect all these parts. This can be applied indifferently to receivers and transmitters. This design includes packaging constraints which make it suitable for industrial IoT applications. Its analysis and optimization are usually based on either full-electromagnetic analysis, e.g., use of finite element method (FEM), Method of Moments (MoM), etc., or on typical RF models [32].

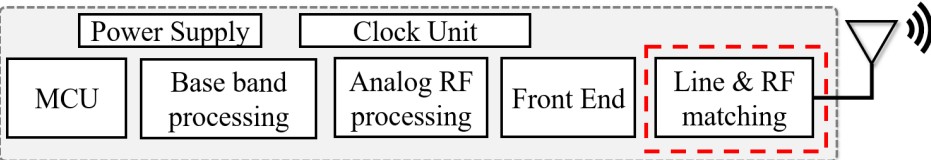

**Figure 1.** Overall architecture of an IoT end point and the focus of this study in red: the design of the line and matching network.

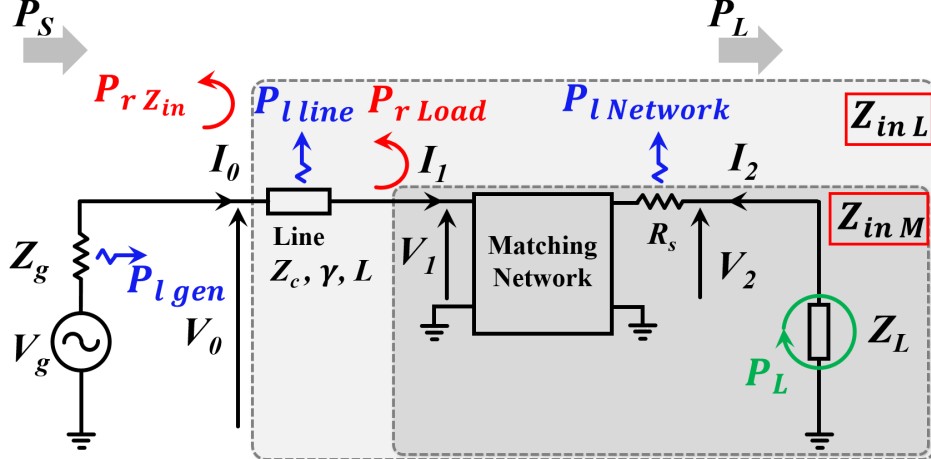

**Figure 2.** Whole transmission chain in an RF system (receiver or transmitter) and associated losses.

In addition to this RF systems design approach, our work within this paper is to present a review and harmonization of different theoretical techniques of impedance matching and to summarise through the presentation of a versatile design framework, based on these well-known theoretical methods such as the transducer gain [41], the transmission line theory [42] or the power waves concept [43]. We highlight that these approaches are equivalent and all give the same expression for the power gain in typical RF systems.

The contribution of this work is the presentation of three analytical, closed-form expressions, based on three theoretical tools, which describe the power delivered to a load in a typical and comprehensive RF system constituted of a generator, a matching network, a transmission line and a load. Thus, the proposed analytical design framework reduces the design computational effort and time.

Hence, in this work, we first propose to harmonize all these theoretical approaches by expressing the power gain of a comprehensive RF system as the one shown in Figure 2 for all of these approaches. This provides system integration engineers with a better understanding of power gain in RF systems, and of how all the different theoretical approaches relate to each other. Then, we use these theoretical formulations to propose a design framework that can be used by any RF engineer or researcher to quickly design and prototype efficient comprehensive RF systems.

This paper is structured as follows: in Section 2, we extend and harmonize the different theoretical approaches. In Section 3, we extend these formulas into a framework to design efficient RF systems. In Section 4, we validate these formulas by comparing them with simulation software, and we use the proposed framework to optimize the design of a Bluetooth RF system. The seminal findings are then summarised in Section 5.

## 2. Formal Expressions of the Power Gain in Typical RF Systems

In typical RF communication systems, a voltage source $V_g$ (electromotive force, e.g., antenna) with an internal impedance $Z_g$ sends a modulated signal to a load $Z_L$ (e.g., IC) through a transmission line of characteristic impedance $Z_c$ and length $L$. In order to reduce reflections, a matching network is usually integrated between the line and the load, as shown in Figure 2. Furthermore, in some applications such as Radio Frequency

Identification (RFID), a serial resistance $R_s$ is added to the matching network in order to increase the operating bandwidth [44]. In this work, we also cover this case and the final under study system's schematic which is represented in Figure 2. Therefore, this work can be applied indifferently for a receiver (in which case the source consists of an antenna for example) or for a transmitter (in which case the load corresponds to an antenna for example).

The aim of RF systems' design is to maximize $P_L$, the delivered power to the load. $P_L$ depends on the ohmic losses of the system and on the return losses due to the mismatching between the parts of the system, which can be reduced by inclusion of a matching network.

In this work, the power $P_L$ is expressed as a function of all the system's design parameters, i.e, $V_g, Z_g, Z_c, \gamma, L, Z_L$, and the matching network components, where $L$ is the length of the line, and $\gamma$ is the propagation constant of the transmission line. This section provides an overview of different theoretical approaches that formulate the power gain within an RF system, and extends them to the system displayed in Figure 2 that includes a transmission line and a matching network. Table 1 provides a comparative summary of the various theoretical approaches that will be addressed in this paper.

**Table 1.** List and comparison of the different power gain formal expressions described in this paper.

| Approach Name Name | Strengths | Weakness of Usual Use Case |
|---|---|---|
| ABCD parameters | Multiplication of matrices that depend on system's impedances | None |
| Transducer gain | Expressed as a function of scattering parameters | Does not include transmission line |
| Transmission lines theory | Simple formula models the effects of the line | Expression depends on unknown parameters ($V_0^i$ in (24)) |
| Power waves | Simple formula | Does not include transmission line nor matching network |

### 2.1. ABCD-Parameters Approach

In this subsection, we describe how ABCD parameters of two-port networks can be used to express the power accepted by the load $Z_L$ in the configuration described in Figure 2. This approach to determine RF systems' power gain will be the reference for the rest of the paper.

The matching network is a two port network, defined by its ABCD-parameters matrix $\begin{bmatrix} A & B \\ C & D \end{bmatrix}$ such that:

$$\begin{bmatrix} V_1 \\ I_1 \end{bmatrix} = \begin{bmatrix} A & B \\ C & D \end{bmatrix} \cdot \begin{bmatrix} V_2 \\ -I_2 \end{bmatrix}. \tag{1}$$

Similarly, the transmission line can also be defined by an ABCD-parameters matrix $[M_{Line}]$ given by:

$$[M_{Line}] = \begin{bmatrix} \cosh(\gamma L) & Z_c \sinh(\gamma L) \\ \frac{1}{Z_c} \sinh(\gamma L) & \cosh(\gamma L) \end{bmatrix} = \begin{bmatrix} M_{11} & M_{12} \\ M_{21} & M_{22} \end{bmatrix} \tag{2}$$

where $L$ is the length of the transmission line and $\gamma$ is the propagation constant of the transmission line, given by $\gamma = \sqrt{(r + jl\omega)(g + jc\omega)} = \alpha + j\beta$, where $r$, $l$, $c$ and $g$ are defined in Figure 3.

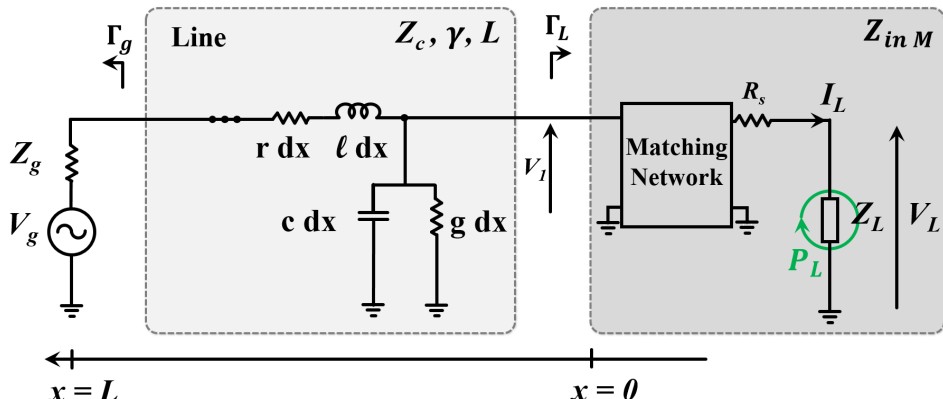

**Figure 3.** System considered in the transmission lines approach.

The active power $P_L$ transmitted to the load $Z_L = R_L + jX_L$ can be computed from the cascade connection between the transmission line, the matching network and the load. Indeed, the relationship between the source's voltage and the load's voltage is given as follows:

$$\begin{bmatrix} V_0 \\ I_0 \end{bmatrix} = [M_{Line}] \cdot \begin{bmatrix} A & B \\ C & D \end{bmatrix} \cdot \begin{bmatrix} V_L \\ -I_2 \end{bmatrix} = \begin{bmatrix} \cosh(\gamma L) & Z_c \sinh(\gamma L) \\ \frac{1}{Z_c} \sinh(\gamma L) & \cosh(\gamma L) \end{bmatrix} \cdot \begin{bmatrix} V_L \\ -I_2 \end{bmatrix} \tag{3}$$

with all voltages and currents defined in Figure 2, such that:

$$\begin{cases} V_g = V_0 + Z_g I_0 \\ I_L = -I_2 = \frac{V_L}{Z_L} \\ V_L = V_2 \end{cases} \tag{4}$$

The active power $P_L$ accepted by the load $Z_L$ can be expressed as follows:

$$P_L = \frac{|V_L|^2}{2} \Re\left[\frac{1}{Z_L}\right] \tag{5}$$

where $\Re[z]$ represents the real part of $z$. Similarly, the power emitted by the source $P_S$ can be expressed as:

$$P_S = \frac{|V_0|^2}{2} \Re\left[\frac{1}{Z_{\text{in L}}}\right] \tag{6}$$

with $V_0 = V_g \frac{Z_{\text{in L}}}{Z_{\text{in L}}+R_g}$ the voltage at the connection point between the output of the source and the impedance $Z_{\text{in L}}$, as shown in Figure 2, where $Z_{\text{in L}}$ is the impedance of the line terminated by the matching network and the load, as seen by the source. $R_g$ is the impedance of the source. After injecting (4) into (3), $Z_{\text{in L}}$ can be computed as follows:

$$Z_{\text{in L}} = \frac{V_0}{I_0} = \frac{M_{11} + \frac{M_{12}}{Z_L}}{M_{21} + \frac{M_{22}}{Z_L}}. \tag{7}$$

Similarly, injecting (4) into (3) leads to the expression of $V_L$ as a function of $V_g$:

$$V_L = \frac{V_g}{M_{11} + Z_g M_{21} + \frac{1}{Z_L}\left(M_{12} + Z_g M_{22}\right)} \tag{8}$$

Hence, the power accepted by the load can be expressed as shown in (9).

$$P_L = \frac{|V_g|^2}{2} \left| \frac{1}{M_{11} + Z_g M_{21} + \frac{1}{Z_L}\left(M_{12} + Z_g M_{22}\right)} \right|^2 \Re\left[\frac{1}{Z_L}\right]. \tag{9}$$

For some generators, $V_g$ is not known, as only the power under 50 $\Omega$ is controllable, in which case it can be interesting to express (9) as a function of $P_s^{max}$ instead of $V_g$, where $P_s^{max}$ is given in (10).

$$P_s^{max} = \frac{1}{4Re[Z_g]} \frac{V_g^2}{2}.$$

(10)

This will lead the power accepted by the load as a function of the maximum power from the source. Based on this expression, we will demonstrate how alternative theoretical approaches converge to a similar expression, thereby demonstrating a robust theoretical design framework for optimal power gain.

### 2.2. Transducer Gain

The transducer gain corresponds to the ratio between the power accepted by a load and the maximum average power available from a source. The transducer gain is usually expressed for a system that is constituted of a source, a matching network and a load, and can be extended to include the transmission line between the matching network and the load. Therefore, in this paper, we extend the expression of the transducer gain to the case of Figure 2 that includes a transmission line.

The demonstration of the transducer gain expression is based on the consideration of travelling waves $a_i$ and $b_i$ for each two port network that constitute the system (the matching network, the transmission line), with $i = 1$ or 2 for the waves that are upstream or downstream the considered 2-port network, respectively. The travelling waves are defined as follows [45]:

$$\begin{cases} a_1 = \frac{V_1^i}{2\sqrt{Re[Z_{0_1}]}} & a_2 = \frac{V_2^i}{2\sqrt{Re[Z_{0_2}]}} \\ b_1 = \frac{V_1^r}{2\sqrt{Re[Z_{0_1}]}} & b_2 = \frac{V_2^r}{2\sqrt{Re[Z_{0_2}]}} \end{cases}$$

(11)

where $Z_{0_1}$ and $Z_{0_2}$ are normalizing impedances of the two ports (1 and 2). $V_1^i$ and $V_1^r$ are the incident and reflected voltage waves amplitudes at port 1 (the same applies for port 2). Indeed, the voltage and the current at each location $x$ of the line is given by the sum of an incident and reflected waves, as shown in [42]:

$$\begin{aligned} V(x) &= V_0^i e^{j\omega t} e^{\gamma(x-L)} + V_0^r e^{j\omega t} e^{-\gamma(x-L)}, \\ I(x) &= I_0^i e^{j\omega t} e^{\gamma(x-L)} + I_0^r e^{j\omega t} e^{-\gamma(x-L)} \end{aligned}$$

(12)

with $V_0^r$ and $V_0^i$ the amplitudes of the reflected and incident wave from the source, respectively. As shown in Figure 3, $x$ is the distance from the matching network, and L is the distance between the matching network and the source.

The S-parameters of the matching network are defined by the matrix $\begin{bmatrix} S_{11} & S_{12} \\ S_{21} & S_{22} \end{bmatrix}$ such that:

$$\begin{cases} b_1 = S_{11}a_1 + S_{12}a_2 \\ b_2 = S_{21}a_1 + S_{22}a_2 \end{cases}$$

(13)

The RF system shown in Figure 2 can be represented as a signal flow graph as shown in Figure 4.

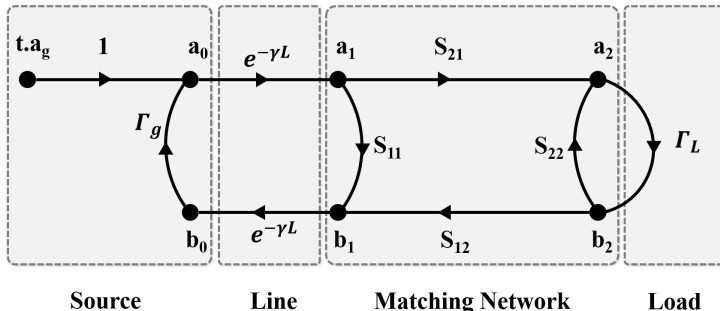

**Figure 4.** Flow graph for the RF system described in Figure 2.

The power accepted by the load $P_L$ is expressed as follows:

$$P_L = |b_2|^2 - |a_2|^2 = |b_2|^2 \left[1 - |\Gamma_L|^2\right] \tag{14}$$

where $\Gamma_L = \frac{Z_L - Z_c}{Z_L + Z_c}$ is the reflection coefficient at the load's end. $Z_c$ is the normalizing impedance that must be taken as equal to the line's characteristic impedance in this particular case. Similarly, the maximum power that the source can provide to a load is given by:

$$P^{max} = |a_0|^2 - |b_0|^2 = \frac{|ta_g|^2}{|1 - |\Gamma_g|^2 \exp^{-2\gamma L}|}\left[1 - |\Gamma_g|^2\right] \tag{15}$$

with $\Gamma_g = \frac{Z_g - Z_c}{Z_g + Z_c}$ the reflection coefficient at the source's end, and $ta_g$ is the wave transmitted by the source after its internal impedance. Then, $b_2$ and $ta_g$ can be expressed by injecting (16) into (13).

$$a_1 = ta_g \exp^{-\gamma L} + \Gamma_g \exp^{-2\gamma L} b_1 \tag{16}$$

The transducer gain is defined as the ratio between $P_L$ (14) and (15) to obtain (17), which corresponds to the comprehensive transducer gain expression for the RF system displayed in Figure 2.

$$G_{t_S} = \frac{|S_{21} \exp^{-\gamma L}|^2 [1 - |\Gamma_L|^2][1 - |\Gamma_g|^2]}{|(1 - \Gamma_g S_{11} \exp^{-2\gamma L})(1 - S_{22}\Gamma_L) - S_{12}S_{21}\Gamma_L\Gamma_g \exp^{-2\gamma L}|^2} \tag{17}$$

Then, we can determine the power accepted by the load by multiplying (17) by $P_s^{max}$ (10), the maximum power available at the source's end. The result is given in (18).

$$P_L = \frac{1}{8}\frac{|S_{21} \exp^{-\gamma L}|^2 [1 - |\Gamma_L|^2]|1 - |\Gamma_g|^2|}{|(1 - \Gamma_g S_{11} \exp^{-2\gamma L})(1 - S_{22}\Gamma_L) - S_{12}S_{21}\Gamma_L\Gamma_g \exp^{-2\gamma L}|^2} Re\left[\frac{1}{Z_g}\right]|V_g|^2 \tag{18}$$

This concludes the expression of the power gain in a comprehensive RF system using the transducer gain approach.

### 2.3. Transmission Lines Approach

Transmission Line theory mostly focuses on the study of the propagation of travelling waves through the transmission line [42]. The system considered in this subsection is the one proposed in Figure 3. The power $P_M$ accepted by the impedance $Z_{in\,M}$ constituted by the matching network terminated by the load, as defined in Figure 3, is given by:

$$P_M = \frac{1}{2} Re[V_L I_L^*] \tag{19}$$

$$= \frac{1}{2} Re \left[ V_0^i \left( e^{-\gamma L} + \frac{V_0^r}{V_0^i} e^{\gamma L} \right) \frac{V_0^{i*}}{Z_c^*} \left( e^{-\gamma L} - \frac{V_0^r}{V_0^i} e^{\gamma L} \right)^* \right] \tag{20}$$

$$= \frac{1}{2} |V_0^i|^2 Re \left[ \frac{1 - |\Gamma_M|^2 + \Gamma_M - \Gamma_M^*}{Z_c^*} \right] e^{-2\alpha L} \tag{21}$$

where $V_1 = V(0)$ is the voltage at the connection between the transmission line and the matching network. $\Gamma_M$ is given by (22) and $Z_c$ is the characteristic impedance of the line given by (23).

$$\Gamma_M = \Gamma(x_{=0}) = \frac{V_0^r}{V_0^i} e^{2\gamma L} = \frac{Z_{\text{in M}} - Z_c}{Z_{\text{in M}} + Z_c} \tag{22}$$

$$Z_c = \frac{V_0^i}{I_0^i} = -\frac{V_0^r}{I_0^r} = \sqrt{\frac{R + jL\omega}{G + jC\omega}}. \tag{23}$$

For a lossless line, (21) can be simplified into:

$$P_L = \frac{1}{2} \frac{|V_0^i|^2}{Z_c} \left[ 1 - |\Gamma_M|^2 \right] e^{-2\alpha L}. \tag{24}$$

However, this well-known expression of $P_L$ depends on the incident wave $V_0^i$ which is unknown most of the time. Therefore, in order to harmonize all theoretical approaches, we propose to extend (24) so it is expressed as a function of known parameters such as $V_g$, the source's voltage. The voltage at the source's ends $V(L)$ as a function of $V_0^i$ is:

$$V(L) = V_g \frac{Z_{\text{in L}}}{Z_{\text{in L}} + Z_g} = V_0^i [1 + \Gamma(L)] = V_0^i \left[ 1 + \Gamma_M e^{-2\gamma L} \right] \tag{25}$$

where $Z_{\text{in L}}$ is the impedance of the load seen by the source, which includes the line and the load, as shown in Figure 3, and is given as follows:

$$Z_{\text{in L}} = Z(L) = Z_c \frac{1 + \Gamma_M e^{-2\gamma L}}{1 - \Gamma_M e^{-2\gamma L}}. \tag{26}$$

Hence, $V_0^i$ as a function of $V_g$ is given as:

$$V_0^i = V_g \frac{Z_c}{Z_c + Z_g} \frac{1}{1 - \Gamma_M \Gamma_g e^{-2\gamma L}} \tag{27}$$

where the reflection coefficients $\Gamma_M = \frac{Z_{\text{in M}} - Z_c}{Z_{\text{in M}} + Z_c}$ and $\Gamma_g = \frac{Z_g - Z_c}{Z_g + Z_c}$ are defined as shown in Figure 3. Based on (27), (21) can be rewritten as shown below:

$$P_M = \frac{1}{2} |V_g|^2 \frac{|Z_c|^2 e^{-2\alpha L}}{|Z_c + Z_g|^2 |1 - \Gamma_L \Gamma_g e^{-2\gamma L}|} Re \left[ \frac{1 - |\Gamma_M|^2 + \Gamma_M - \Gamma_M^*}{Z_c^*} \right] \tag{28}$$

This can be further simplified by noticing that $Re \left[ \frac{1 - |\Gamma_M|^2 + \Gamma_M - \Gamma_M^*}{Z_c^*} \right]$ is equal to $\frac{4 Re[Z_{\text{in M}}]}{|Z_c + Z_{\text{in M}}|^2}$.

This simplification leads to (29) that expresses the active power accepted by a load $Z_{\text{in M}}$ that is connected to a source of electromotive force $V_g$ through a line of characteristic impedance $Z_c$, with a propagation constant $\gamma$ and a length $L$:

$$P_M = 2|V_g|^2 \frac{|Z_c|^2 Re\{Z_{\text{in M}}\} e^{-2\alpha L}}{|(Z_{\text{in M}} + Z_c)(Z_g + Z_c)e^{\gamma L} - (Z_{\text{in M}} - Z_c)(Z_g - Z_c)e^{-\gamma L}|^2} \tag{29}$$

The same result could have been obtained by seeing that $P_M = \frac{1}{2}|V(0)|^2 Re\left[\frac{1}{Z^*_{\text{in M}}}\right]$, where $V(0) = V^i_0(1 + \Gamma_M)$, and $V^i_0$ is given by (27). Then, noticing that $|1 + \Gamma_M|^2 Re\left(\frac{1}{Z^*_{\text{in M}}}\right) = \frac{4Re[Z_{\text{in M}}]}{|Z_c + Z_{\text{in M}}|^2}$ leads to (29).

This expression of $P_M$ is now expressed as a function of the source's voltage, while capturing the mismatch between the source and the terminated line impedance. It also includes the losses within the line and the mismatch between the line and the load. However, the matching network has not been included into the equation yet.

To integrate the active power losses due the matching network into (29), we remove the losses from the resistive components of the matching network. This is important especially for applications as Wireless Power and Data Transfer (WPDT) where the quality factor of the matching network needs to be lowered by a resistance $R_s$ in order to increase its bandwidth. Hence, neglecting the ohmic losses in other components of the matching network, the active power accepted by the load $P_L$ is equal to the power $P_M$ transmitted to $Z_{\text{in M}}$, subtracted by the active power dissipated in this potential matching network's resistance $R_s$. This leads to:

$$P_L = P_M \frac{Re\{Z_L\}}{Re\{Z_L\} + R_s} \tag{30}$$

$$= \frac{2|V_g|^2|Z_c|^2 Re\{Z_{\text{in M}}\}e^{-2\alpha L}}{|(Z_{\text{in M}} + Z_c)(Z_g + Z_c)e^{\gamma L} - (Z_{\text{in M}} - Z_c)(Z_g - Z_c)e^{-\gamma L}|^2} \frac{Re\{Z_L\}}{Re\{Z_L\} + R_s} \tag{31}$$

Hence, this subsection has provided a reformulation and extension of the transmission line theory that demonstrates that travelling waves can also be used to formulate the power gain of a comprehensive RF system. More importantly, this formulation is not depending on the amplitude of incident waves, but is now expressed as a function of controllable parameters, such as the voltage of the source $V_g$. In the next subsection, we describe another theoretical approach based on power waves.

### 2.4. Power Waves

Following the development from the transmission line theory, which is based on travelling waves, Kurokawa has described the concept of power waves in [43]. The system considered by Kurokawa consists of a load $Z_L$ connected directly to the source. We will extend this approach so it can be applied to a more comprehensive RF system, as the one shown in Figure 2. Kurokawa defined the power waves $a_p$ and $b_p$ as the incident and reflected power waves (respectively) on a two-port network. Considering the two-port network defined by $Z_{\text{in L}}$ comprised of the transmission line connected to the matching network and the load, as displayed in Figure 2, we can express $a_p$ and $b_p$ as follows:

$$a_p = \frac{V_0 + Z_0 I_0}{2\sqrt{Re[Z_0]}}, \quad b_p = \frac{V_0 - Z^*_0 I_0}{2\sqrt{Re[Z_0]}} \tag{32}$$

where $a_p$ is the incident power wave sent by the source and received by the 2-port network constituted by the line terminated by the matching network and the load. $b_p$ is the power wave reflected by this 2-port network, $V_0$ and $I_0$ are the 2-port network's voltage and current as defined in Figure 2, and $Z_0$ is the normalizing impedance corresponding to the source impedance $Z_g$ in this case [45].

The power $P_{\text{in L}}$ accepted by the 2-port network corresponding to $Z_{\text{in L}}$ can be expressed by:

$$P_{\text{in L}} = \frac{1}{2}Re[V_0 I^*_0] = \frac{1}{2}\left(|a_p|^2 - |b_p|^2\right) = \frac{R_{\text{in L}_p} V^2_g}{2|Z_{\text{in L}} + Z_g|^2}. \tag{33}$$

where $R_{\text{in L}_p}$ is the resistive part of $Z_{\text{in L}}$ expressed in parallel, and $V_L = V_g - Z_g I_L$. In the case of perfect matching between the 2-port network and the source ($Z_{\text{in L}} = Z^*_g$), the

power accepted by the network is maximal and is given by (10). Hence, as introduced in [46], (33) can be expressed as follows:

$$P_{\text{in L}} = P_s^{max} \frac{4 Re[Z_{\text{in L}}] Re[Z_g]}{|Z_{\text{in L}} + Z_g|^2} = P_s^{max}(1 - |\Gamma_{p_{\text{in L}}}|^2) \tag{34}$$

where $\Gamma_{p_{\text{in L}}} = \dfrac{b_p}{a_p} = \dfrac{Z_{\text{in L}} - Z_g^*}{Z_{\text{in L}} + Z_g}$ is the power waves reflection coefficient, not to be confused with $\Gamma_L$ expressed in the Transmission line theory subsection that applies to travelling waves (Section 2.3). This expression reflects the losses due to reflection between the source's impedance and the impedance $Z_{\text{in L}}$. In order to express the power received by the load $Z_L$, we must now consider the losses that are due to the transmission line and the matching network. First, we can include the reflection losses between the transmission line and the impedance $Z_{\text{in M}}$ by formulating $Z_{\text{in L}}$ as follows:

$$Z_{\text{in L}} = Z_c \frac{Z_{\text{in M}} + Z_c \tanh \gamma L}{Z_c + Z_{\text{in M}} \tanh \gamma L}. \tag{35}$$

If we also include the active power losses in the matching network (considering only the losses in the resistance $R_s$), as mentioned in Section 2.3, we can express the active power accepted by the load $P_L$ as follows:

$$P_L = P_{Z_{inM}} \frac{Re[Z_L]}{Re[Z_L + R_s]} \approx P_s^{max} \left(1 - |\Gamma_{p_{\text{in L}}}|^2\right) \frac{Re[Z_L]}{Re[Z_L + R_s]} \tag{36}$$

where $\Gamma_{p_{\text{in L}}}$ is expressed using $Z_{\text{in L}}$ given by (35). Hence, this approach can also provide an expression of the active power accepted by a load connected to a source through a matching network and a transmission line. However, unlike all the other formulation displayed so far, it does not include the losses inside the transmission line, which explains why (36) is only an approximation. Hence, unlike (9), (18) and (31), (36) should only be used in the case of lossless transmission lines.

This section summarized and extended different theoretical approaches that can all be used to formally express the power received by the load in a typical RF system as the one displayed in Figure 2. In the next section, we will study how these formal expressions can be used to optimize the design of RF systems by an analytical approach as an alternative to RF simulation software.

## 3. Optimal Design of RF Systems Using an Analytical Approach

As presented in the previous section, (9), (18) and (31) give three different ways to formally express the power accepted by a load $Z_L$ that is connected to a source $V_g$ through a transmission line and a matching network. In this section, we propose to extend the formulas described above into a framework that one can use to completely design an efficient RF system by determining the components parameters that will optimize indifferently (9), (18) or (31). This framework aims to find the optimal line geometry, matching network components and source and load impedances that maximize the system's efficiency. Hence, this framework is based on an optimization of (9), (18) and (31). Before describing the framework, it is necessary to express $\gamma$ and $Z_c$, the propagation constant and the characteristic impedance of the transmission line and $Z_{\text{in M}}$ the impedance of the matching network terminated by the load as a function of all the design parameters (geometry, components). This will allow us to obtain a comprehensive expression of the power accepted by the load as a function of all the design parameters.

### 3.1. Integration of the Transmission Line Parameters

The transmission line characteristic impedance $Z_c$ and its propagation constant $\gamma$ can both be expressed as functions of the line's geometric parameters based on existing formulas, as shown in [42,47]. Therefore, for a microstrip line, (9), (18) or (31) can be expressed as explicit formulas of the line's copper width $w$, the substrate height $h$, and the length of the line $L$.

In this section, we present the case of a straight microstrip line, which is the most common configuration used by RF engineers. Hence, for a microstrip line as defined in Figure 5, the propagation constant $\gamma$ can be defined by (37), while $Z_c$ is defined by (38).

$$\gamma = \alpha + j\beta = \alpha + j2\pi f \frac{\sqrt{\varepsilon_{eff}}}{c_0} \tag{37}$$

with $c_0$ the speed of light in vacuum, and $\alpha$ is given by [48].

$$Z_c = \begin{cases} \dfrac{60}{\sqrt{\varepsilon_{eff}}} \ln(\dfrac{8h}{w} + \dfrac{w}{h}), & \text{if } \frac{w}{h} < 1 \\ \dfrac{1}{\sqrt{\varepsilon_{eff}}} \dfrac{120\pi}{\dfrac{w}{h} + 1.393 + 0.667 \ln(\dfrac{w}{h} + 1.444)}, & \text{if } \frac{w}{h} \geq 1 \end{cases} \tag{38}$$

where $\varepsilon_{eff}$ is the effective relative permittivity of the substrate, defined by (39) [49], and $w$ and $h$ are the width of the microstrip line and the thickness of the substrate respectively, as shown in Figure 5:

$$\varepsilon_{eff} = \frac{(\varepsilon_r + 1)}{2} + \frac{(\varepsilon_r - 1)}{2\sqrt{1 + 12\dfrac{h}{w}}} \tag{39}$$

Hence, replacing $\gamma$ and $Z_c$ by (37) and (38), respectively, in (9), (18) or (31).

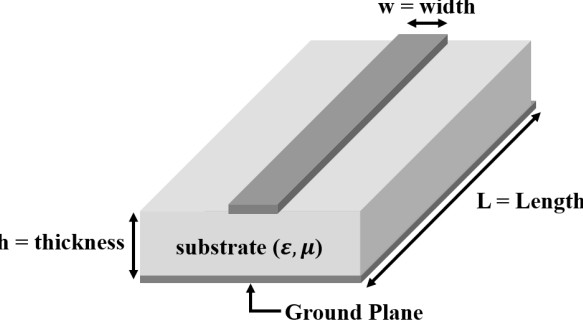

**Figure 5.** Description of the considered microstrip as an example, where the geometrical parameters to be optimized have been highlighted (width, length and thickness).

### 3.2. Integration of Matching Network Parameters

In this subsection, we express $Z_{\text{in M}}$ as a function of the matching network design parameters so all expressions of $P_L$ ((9), (18) or (31)) can be fully expressed as a function of explicit design parameters. The objective of a matching network is to minimize the reflection between the transmission line and the load. Different architectures can be chosen for matching networks (L-shape, T-shape, ...), depending on the application, space and cost. For each of these architectures or topology, there is a formula to express $Z_{\text{in M}}$ as a function of the load and the matching network components' impedances [50]. As an example, using the ABCD parameters as expressed in (41) for a T-shaped matching network such as the one shown in Figure 6, we can determine the input impedance $Z_{\text{in M}}$ seen by the transmission line using (40):

$$Z_{inM} = \frac{A \cdot Z_L + B}{C \cdot Z_L + D} \tag{40}$$

where $A, B, C$ and $D$ are determined as follows for a T-shaped matching network:

$$\begin{cases} A = 1 + \frac{Z_a}{Z_p} & \qquad C = \frac{1}{Z_p} \\ B = Z_a + Z_b + \frac{Z_a Z_b}{Z_p} & \qquad D = 1 + \frac{Z_b}{Z_p} \end{cases} \tag{41}$$

where the impedances $Z_a, Z_b$ and $Z_p$ are the impedances of the components located as defined in Figure 6. For example, if $Z_a$ corresponds to an inductance $L_a$, then $Z_a = jL_a\omega$.

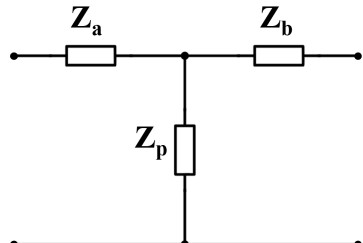

**Figure 6.** T-Shaped matching network corresponding to (41).

Finally, in real applications, the matching network's components are soldered between each other, which necessitates to include small lines between each component into (9), (18) or (31). This can be done by integrating these small transmission lines into the matching network impedances by replacing $Z_a, Z_b, Z_p$, and $Z_L$ in the computation of $Z_{\text{in M}}$ by $Z_{a_{\text{line}}}, Z_{b_{\text{line}}}, Z_{p_{\text{line}}}$, and $Z_{L_{\text{line}}}$ given as follows [42]:

$$Z_{a_{\text{line}}} = Z_{c_{\text{line}}} \frac{Z_a + Z_{c_{\text{line}}} \tanh d\gamma_{\text{line}}}{Z_{c_{\text{line}}} + Z_a \tanh d\gamma_{\text{line}}} \tag{42}$$

where $d$ is the length of the small line added before the component of impedance $Z_a$ for soldering, $Z_{c_{line}}$ and $\gamma_{line}$ (computed using (38) and (37) respectively) correspond to its characteristic impedance and propagation constant, respectively, and $Z_{a_{line}}$ is the new impedance of the component $Z_a$ including this small line for soldering. The same applies for $Z_b, Z_p$, and $Z_L$ by replacing $a$ in (42) by $b, p$ and $L$, respectively. Hence, (9), (18) or (31) determine formally the active power $P_L$ accepted by the load as functions of all design parameters. Therefore, we can now implement and optimize one of these expressions in order to determine the best parameters that maximize the active power accepted by the load.

*3.3. RF Design Framework*

From (9), (18) or (31), it is now possible to find the transmission line geometry and the matching network components and topology that will optimize $P_L$, as shown in the optimization problem described below. The proposed framework is displayed in Figure 7.

The first steps consist of defining the type of the line, the architecture of the matching network, and the variables that should be optimized, as described below. Then, the last two steps of the design framework consist in an optimization problem, where the objective is to maximize (9), (18) or (31) by changing the line geometry and matching network components values, while meeting the constraints such that all distances must fit into the packaging and that the system's gain must satisfy some bandwidth requirements. Given the optimization functions, the optimization problem is a Nonlinear Problem (NLP) that can be implemented in some spreadsheet or optimization software (MATLAB, R, GAMS or Excel), and optimized using an appropriate nonlinear solver as population based optimization algorithms. The expression of this problem for microstrip lines and T-shaped matching network is given below, where the optimization aims to find the best optimization

variables (line geometry and matching network components) that maximize the system's power efficiency:

$$\underset{h,\,w,\,L,\,\rho,\,Z_a,\,Z_b,\,Z_c}{\text{maximize}} \quad P_L(\omega_0)\ (9),\ (18)\ or\ (31)$$

$$\text{subject to} \quad \text{Distances} \geq 0$$
$$\text{Packaging constraints}$$
$$P_L(\omega) \geq \eta\ \forall \omega \in [\omega_1, \omega_2].$$

(43)

where $\eta$ is the threshold above which the RF system gain should be for angular frequencies $\omega$ between angular frequencies $\omega_1$ and $\omega_2$ in order to maintain communication performance requirements, and $\omega_0$ is the carrier frequency. Other frequency requirements could also be added to this constraint. The source and load impedances $Z_g$ and $Z_L$ respectively can also be included in the optimization variables in order to find the best impedance targets for the antenna and the system IC. In the case of nonlinear impedance IC, we can model the impedance as a variable that depends on the frequency or other parameters using a look-up table that can be derived from Load Pull measurement techniques [51,52] or custom methods as shown in [53] for RFID measurements. Although this makes the optimisation problem more complex to solve, population based algorithms are well fitted to find global extremum for such nonlinear problems. The solution of this problem corresponds to the optimal line geometry and components values that will maximize the efficiency of the RF system. Finally, an iterative process can be done on the type of line and matching network geometry in order to find the optimal system that will reach the desired overall efficiency.

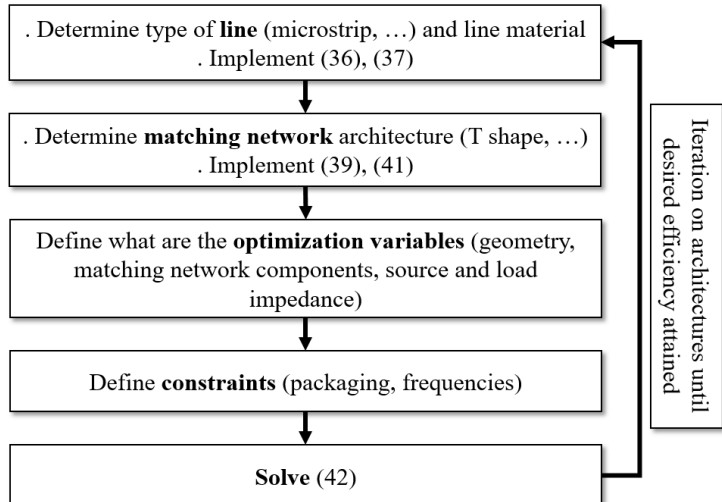

**Figure 7.** Framework for efficient design of RF systems based on theoretical approaches.

In this study, this problem was optimized using population-based evolutionary algorithm available in Excel. Although evolutionary algorithms often take considerable computing time to converge, they are best fitted to find global extrema in such constrained non-convex optimization problems [54], although other algorithms such as pattern search can also achieve similar results. CPU processing time for solving such an optimization problem without bandwidth requirement was below one minute for all the simulations realized in this study.

## 4. Validation and Implementation of the Design Framework

In this section, we first validate the theoretical formulas from Section 2 by comparing the measured accepted power by a 50 $\Omega$ load with the computed accepted power using (9), (18) and (31). Once the formulas from the different theories are validated, we will use the proposed framework for the design of a typical RF system and compare the results with RF simulation software.

### 4.1. Validation of Theoretical Approaches

To validate the accuracy of all the theoretical approaches, we propose to compare them with experimental measurement results as measured by a Vector Network Analyser (VNA). To do so, the circuit presented in Figure 8 is fabricated. It is a single microstrip line connected to a matching network on a low-cost FR-4 substrate. The FR-4 characteristics and dimensions of the line are shown in Table 2. The matching network consists of a T-shape matching network with CMS lumped components as mentioned in Table 2, with a 1608 Metric case. In order to measure the power accepted by the load, it was chosen to replace the resistive part of the load by one VNA port's internal impedance (50 Ω). Thus, the measured active power received by the load is given by (44).

$$P_{L_{\text{measured}}} = |S_{21}|^2 P_{\text{max VNA}} \tag{44}$$

where $P_{\text{max VNA}}$ corresponds to the power sent by the VNA through port 1, which is obtained by connecting a 50 Ω load to port 1.

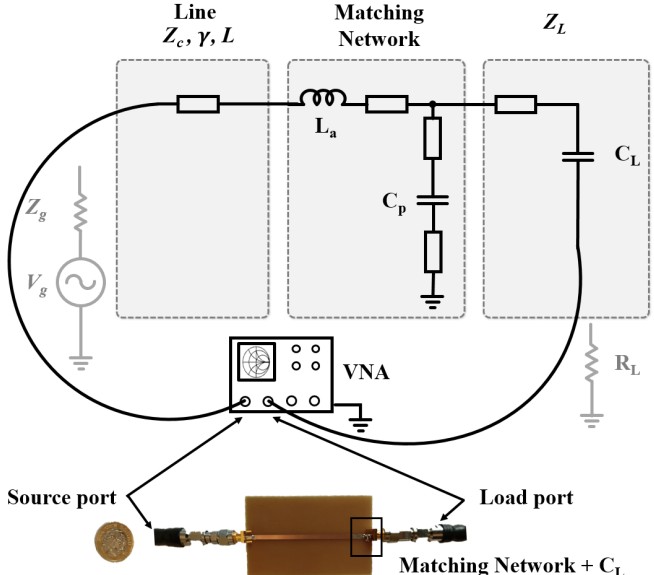

**Figure 8.** Test bench used for validation of (9), (18) and (31).

**Table 2.** Parameters used for the measurement setup.

| Parameter | Value | Unit | Parameter | Value | Unit |
|---|---|---|---|---|---|
| $Z_g$ | 50 | Ω | $d$ | 1.5 | mm |
| $Re\{Z_L\}$ | 50 | Ω | $L_a$ | 1 | nH |
| $w$ | 2.3 | mm | $C_p$ | 1 | pF |
| $h$ | 1.6 | mm | $C_L$ | 10 | pF |
| $L$ | 67 | mm | $\varepsilon_r$ | 4.4 | |

In addition to the experimental measurement, we also compare the accepted power with a simulation realised with an Advanced Design System (ADS). Figure 9 displays the comparison of the obtained accepted power gains for the different methods. The power gain is defined as the ratio between the power accepted by the load and the maximum power from the source.

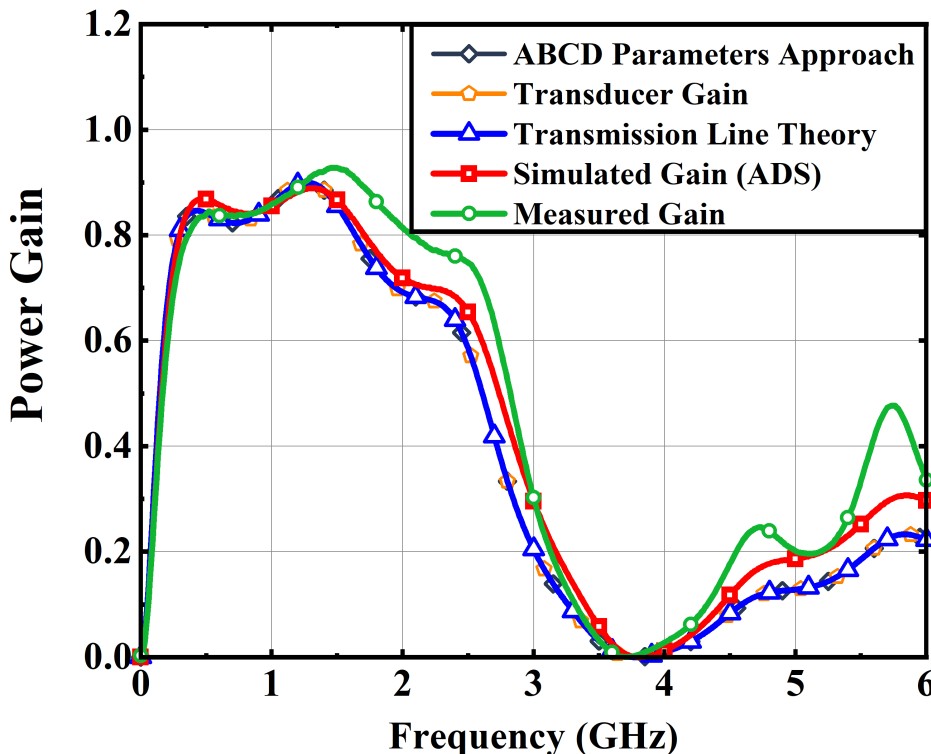

**Figure 9.** Comparison of gain obtained during measurements, simulation on ADS and with formulas (9), (18) and (31) that are all coincident.

As shown in Figure 9, the results from the theoretical approaches proposed in (9), (18) and (31) are all coincident and capture well the evolution of the measured power gain as they follow the waves that correspond to constructive or destructive interferences between the incident and reflected waves at the load's location. The difference in absolute value between the experimental measurement and the theoretical approaches is below 30% for the 0–4 GHz range, whereas most of the errors are due to the fact that paths between the line and the ground plane are not modelled in the proposed formulas, as well as losses in connectors and connecting cables. However, one can comment that the theoretical approaches give similar results to ADS simulation (with a percentage error of 15% below 4 GHz), which is of great interest as it includes most of recent IoT applications, such as Bluetooth, Wifi, LoRa and 5G. Hence, (9), (18) or (31) are easily implementable in Excel spreadsheet software by any RF engineer or researcher, and can then be optimized very quickly in order to find the optimal line geometric parameters and matching network components that will maximize the power received by the load. Given the accuracy of the approach, this allows any RF engineer to design efficient simple RF systems without the need for expensive RF simulation software or tutorial on proprietary software. Given the agreement between simulations and experimental results that include the discontinuity and the parasitic effects, we have not taken into account these phenomena further in our design framework because they seem to have a low impact in our case under testing. In the next subsection, we will use the proposed framework to design an efficient Bluetooth system for smart glasses.

### 4.2. Implementation of the Proposed Design Framework

In order to show the advantages of the proposed analytical framework, we propose to integrate a Bluetooth chip into an IoT system for smart glasses. The integration consists of designing a transmission line that connects the Bluetooth chip (the source) with a matching network and the antenna (the load). The source has a configurable impedance of 75 $\Omega$, and the antenna (which corresponds to the load in the previous sections) is a half wave antenna modelled as a 72 $\Omega$ resistance (at 2.45 GHz) with a serial capacitance. The line

has a geometric constraint to be greater than 3 cm and less than 5 cm, as it is the case in most smart glasses to fit into the branches [55,56]. In addition, the manufacturing process did not allow for achieving a characteristic impedance of 75 $\Omega$ for the transmission line. The framework described in Figure 7 was used to design the system. For the last step of Figure 7, (31) was implemented in a spreadsheet solver (Excel with the integrated solver add-in) and optimized over the design variables that are the line's width $w$, its length $L$ and the matching network components (an inductance $L_a$ and a capacitance $C_p$). Although we limited this use case to a matching network of second order, the framework can indifferently be used for larger orders' matching networks, in which case the number of variables would increase with the number of components to size. Values of the different parameters obtained for this very specific use case are displayed in Table 3. The output of the optimization process is a new line width and matching network components' values that maximize the power accepted by the BlueTooth antenna at the specified frequency. Using these components' values, we then computed and compared the accepted power by the load for a large frequency range (7 GHz) using the formula from the transmission line theory (31), the power waves approach that does not include the line losses (36), the transducer gain formula (18), the ABCD parameters approach (9), and also the result from the transducer gain when the transmission line is not considered. Finally, we also simulated the corresponding system on ADS. The resulting accepted power curves are displayed in Figure 10, whereas the final architecture and the S-parameters of the matching filter are shown in Figure 11.

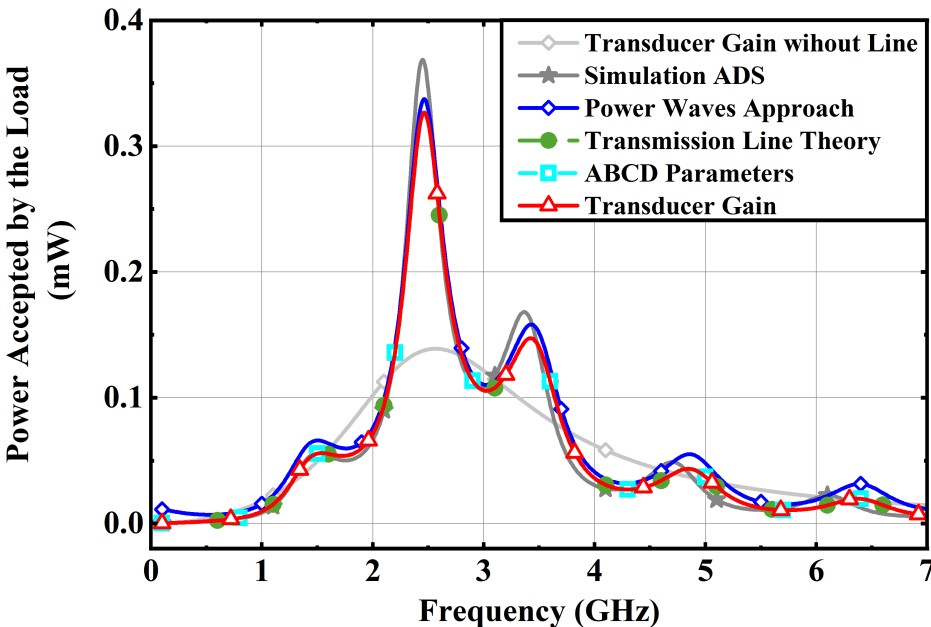

**Figure 10.** Comparison of the power accepted by the load for different approaches.

**Table 3.** Parameters used for the Bluetooth system design.

| Parameter | Value | Unit | Parameter | Value | Unit |
|---|---|---|---|---|---|
| $Z_g$ | 75 | $\Omega$ | $L_a$ | 5.7 | nH |
| $Re\{Z_L\}$ | 72 | $\Omega$ | $C_p$ | 0.6 | pF |
| $w$ | 2.1 | mm | $C_L$ | 0.1 | pF |
| $h$ | 1.6 | mm | $\varepsilon_r$ | 9.6 | |
| $L$ | 37 | mm | | | |

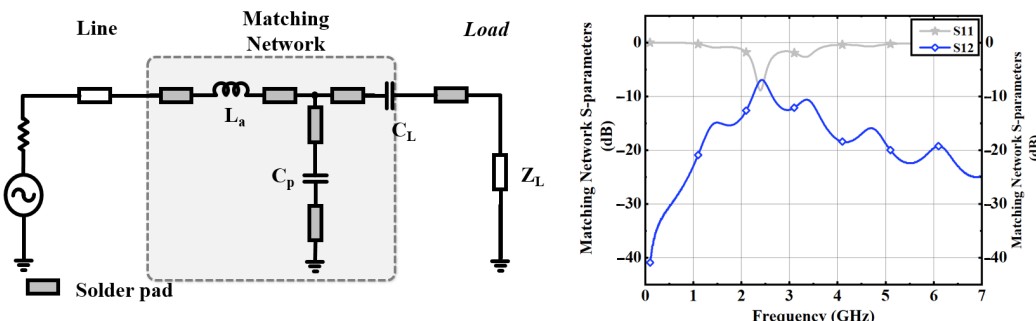

**Figure 11.** Final design for the Bluetooth system design meeting the geometric constraints, and associated S-parameters.

The results shown in Figure 10 demonstrate the performance of the proposed framework, due to its ability to capture the evolution of the accepted power with the frequency as ADS does. We can remark that the power waves approach (36) gives slightly different results than other approaches, which is due to the fact that it does not include the losses in the transmission line. Although it was demonstrated that all other theoretical approaches are equivalent, the authors recommend to use the ABCD parameters approach (9) when optimizing the design of RF systems because it is the simplest one to implement. However, if the framework works well for simple designs with a source, a line, and a matching network, it does not replace RF software simulations for more complex systems or more complex lines' geometries. Hence, it is of great interest for teaching purposes, or for quick proof of concept when there is no access to RF simulation software.

Finally, (9), (18) and (31) are useful for rapid sensitivity study over one or several design parameters, using only a spreadsheet (as excel with the add-in solver) instead of RF simulation software, which can be interesting for laboratory tutorials. For example, we can fix all the system's parameters except the length of the micro-strip line $L$, and study its impact on the accepted power, as one can do using the tune option in ADS software. This was done for the design proposed above, where the maximal accepted power was obtained for a microstrip length of 13 and 37 mm either with the theoretical formulas (the ABCD parameters were used) or with ADS software, as shown in Figure 12. The low resistivity of the material used explain the small reduction in accepted power with the length increase.

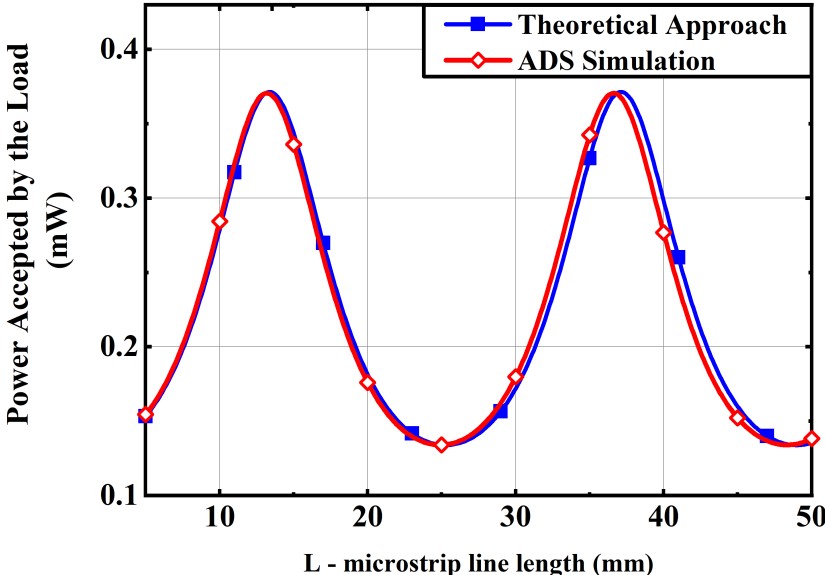

**Figure 12.** Impact of the microstrip length $L$ on the accepted power $P_L$.

In this section, it was shown that the proposed framework computes accurately the power accepted by a load or an antenna of a typical RF system corresponding to the architecture shown in Figure 2. The maximum percentage error as compared with experimental measurement is below 30%, whereas the maximum error value when compared with ADS simulations is below 15% for frequencies lower than 4 GHz. Given the simplicity of its implementation, it can easily be used by any electronic engineer to integrate efficient IC into bespoke IoT systems. Furthermore, this optimization of formal expressions of $P_L$ can be done at a lower cost than using RF finite elements simulations, and does not require any prior knowledge in RF computer aided design (CAD) software.

## 5. Conclusions

Energy efficient designs are fundamental to the continued growth of IoT applications, and should be applied to the whole IoT systems, including the design of the transmission lines and matching network. However, such efficient designs for bespoke RF systems with strong packaging constraints are usually achieved through RF simulations software which restrains the access to efficient system design. Hence, this paper described a framework to design efficient RF systems that is based on theoretical approaches instead of RF simulations, and that achieves similar results to RF CAD software. It can be used as a replacement approach or as a validation method of design achieved through RF simulations. First, we extended and harmonized different RF theoretical approaches in order to express the power gain of a comprehensive RF system. The transducer gain approach and transmission line theory were extended so they can be used to determine the power gain of any RF system constituted by a source, a transmission line, a matching network and a given load. Furthermore, it was highlighted that these formal expressions give similar results to RF simulation software, even when we include the soldering of components. Then, we used these validated expressions to propose the aforementioned analytical framework to design efficient RF systems. This framework can be used to quickly determine a matching network, a transmission line geometry, and the source and load impedances that maximize the power accepted by the system's load (IC) or antenna. Simulation and experimental results showed that the proposed framework provides similar solutions as RF simulation software for frequencies below 4 GHz, which makes it suitable for IoT devices' design that use communication standards such as Wifi, Bluetooth and low and mid-band 5G. Given the underpinning role of IoT to future business models and disruptive technologies, our design framework/methodology provides a strategic insight and enabler for efficient IoT design and implementation.

**Funding:** This research was funded by Engineering and Physical Sciences Research Council, Grant No. EP/L014998/1.

**Conflicts of Interest:** The authors declare no conflict of interest.

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
