# Peer review of "Internet of Things: A Review on Theory Based Impedance Matching Techniques for Energy Efficient RF Systems"

_jlpea, doi:10.3390/jlpea11020016_

Round 1
Reviewer 1 Report
The authors presented a systematic procedure for designing an RF wireless power transfer circuit. The simulated results of using the proposed framework and the results using ADS agree well. The good agreement validates the proposed framework. However, the proposed design flow in Fig. 7 dose not seem to show great innovations. The equations used are well known and thus a good agreement is obvious. Also, the iterative procedure is straightforward. The discontinuity and parasitic effect may be the most significant different for circuit model and EM simulation. It would be better that the authors can explain how the iterative procedure can handle the discontinuity and parasitic effect.
Reviewer 2 Report
This paper review the theory based impedance matching techniques for energy efficient RF systems in IOT. It is of interested to readers and peers working in this field. Some moderate revisions are required.
After ref. 27, the RF CMOS RF optical transmitters like doi: 10.1109/JSEN.2016.2582840 could be mentioned; in Fig. 5, for the length, width, and thickness of the proposed device structure, these parameters should be indicated in the figure, also is there any way to optimize the values should be further clarified; in Fig. 9, shapes of each case are quite similar, implying the correlation should be almost linear, such an almost linearity should be further indicated by adding a new figure, as to show the theory could match the approach, the simulated could be in good agreement with measured reasonably well, also the gain of zero always occurs at~3.6, first why there is a bottom, second why it occurs at ~3.6, third is there any way to optimize the expected frequency to other values rather than ~3.6; in Fig. 11, the curve is almost symmetric, what the advantage of such characteristics should be further clarified.

Author Response
We deeply thank the reviewer for his careful reading and relevant comments.
Here are our responses attached to this form. Please see the attachment.
Kind regards,

Reviewer 3 Report
The article provides an overview as well as various analytical formulas useful for the design of matching networks. These concepts are not new, but I guess they can be useful for the practicioner. The work also provides an implementation example.
Nevertheless, the authors should mention in their work that linear matching network design has always been done by means of the Smith Chart techniques, at least for simple implementations (e.g., lumped elements or elementary transmission lines elements), without involving complex optimization algorithms such as proposed here (e.g., Sec. 3.3). Indeed, when the geometries become more complex, optimization algorithms could be used.
As far as I understand, the authors describe the matching problem only in the linear case. However, when introducing this topic, it should be (at least) mentioned the problem of matching in the presence of active (nonlinear) components, as usually happens in the transmitter/receiver chain. Please clarify these aspects. Note that in the nonlinear case there is not such an analytic theory to be used, but load-pull measurements (A. Ferrero and M. Pirola, 'Harmonic Load-Pull Techniques: An Overview of Modern Systems,' IEEE Microwave Mag. 2013) are used. Note that broadband matching, which usually implements high-order designs and with many reactive elements, requires special special active load-pull measurements for the nonlinear case (A. M. Angelotti, et al. 'Wideband Active Load–Pull by Device Output Match Compensation Using a Vector Network Analyzer,' IEEE Trans. Microwave Theory Techn., Jan. 2021).
- I would suggest to avoid phrases like (line 149) 'The transducer gain is usually expressed for a system that is constituted of a source, a matching network and a load, but to the authors best knowledge, it has never been extended to include the transmission line between the matching network and the load.' In fact, what the authors propose is quite a basic calculus in the practice of matching network design.
- In my opinion, the initial part of the Introduction is too general and does not focus on the actual content of the article. Therefore, I'd suggest to cut much of the text before line 56. I don't really see such unique link with IoT: these are basic aspects of microwave circuit theory.
- Another aspect is that sometimes the authors make reference to a transmitter configuration, some other times to a receiver configuration (e.g. Fig. 2). Also, the caption of Fig. 2 is a bit misleading, as the whole transmission chain is reported, not only the 'power losses'.
- In the expression in terms of ABCD-parameters (Equation 3), why not including the parameters for the line, instead compacting everything in Mxx coefficients?
- In Eq. 11, please include the expressions for V_1 and V_2 voltage waves.
- Regarding the optimization algorithms for non-convex problems with well defined boundaries for the variables, the authors should also mention the so called 'direct' optimization methods (pattern search, simplex, etc.).
- Is there a target application for the test circuit described in Sec. 4.1? At which center frequency did you optimize the matching network? Is it a broadband optimization across a certain bandwidth? At line 279: a 30% error is a huge one. How did you quantify this across the frequency? Please note that 5G is mostly centered at frequencies above 4 GHz (and in the millimeter wave frequencies).
- In Sec. 4.2, it is not immediately clear what are the variables used within the optimization, and also the overall optimization procedure. A flow chart would help the discussion. Also, what is the order of the matching network designed? Beyond showing the delivered power, the authors should provide the S-parameters of the designed matching network and the final schematic.
Author Response
Please see the attachment.
We would thank the editor and all reviewers who provided comments for their very careful reading of the manuscript and their comments, which we believe have helped substantially to improve the paper.
We have considered and addressed each of their comments very carefully, and revised the paper with revised structure, and additional references.
In this document, we provide detailed replies on how we considered and addressed each comment.

Round 2
Reviewer 2 Report
All the issues concerned are addressed appropriately. It is recommended to be published as it is.
Reviewer 3 Report
I thank the authors for addressing my comments.